# Ethnicity and trends in pediatric specialty care clinic attendance at an academic medical center in the rural southeastern US

Philip Maness[1], Dmitry Tumin[2], Rushina Cholera[3], David N. Collier[2], Luisa Bonilla-Hernandez [4], Suzanne Lazorick [2,5] *

1 Brody School of Medicine, East Carolina University, Greenville, North Carolina, United States of America, 2 Department of Pediatrics, Brody School of Medicine, East Carolina University, Greenville, North Carolina, United States of America, 3 Department of Pediatrics, Duke University School of Medicine, Durham, North Carolina, United States of America, 4 Pediatrics Residency Program, Vidant Medical Center and East Carolina University, Greenville, North Carolina, United States of America, 5 Department of Public Health, Brody School of Medicine, East Carolina University, Greenville, North Carolina, United States of America

* lazoricks@ecu.edu

**Data Availability Statement:** A file of de-identified data has been made available and uploaded as a

## Abstract

Following the 2016 US Presidential election, immigration enforcement became more aggressive, with variation by state and region depending on local policies and sentiment. Increases in enforcement created an environment of risk for decreased use of health care services among especially among Latino families. of Hispanic ethnicity and/or from Latin American origin (as a group subsequently referred to as Latino). For Latino children with chronic health conditions, avoidance of routine health care can result in significant negative health consequences such as disease progression, avoidable use of acute health care services, and overall increased costs of care. To investigate for changes in visit attendance during the periods before and since increased immigration enforcement, we extracted data on children followed by subspecialty clinics of one healthcare system in the US state of North Carolina during 2015–2019. For each patient, we calculated the proportion of cancelled visits and no-show visits out of all scheduled visits during the 2016–2019 follow-up period. We compared patient characteristics (at the 2015 baseline) according to whether they cancelled or did not show to any visits in subsequent years by clinic and patient factors, including ethnicity. Data were analyzed using multinomial logistic regression of attendance at each visit, including an interaction between visit year and patient ethnicity. Among 852 children 1 to 17 years of age (111 of Latino ethnicity), visit no-show was more common among Latino patients, compared to non-Latino White patients; while visit cancellation was more common among non-Latino White patients, compared to Latino patients. There was no significant interaction between ethnicity and trends in visit no-show or cancellation. Although differences in pediatric specialty clinic visit attendance by patient ethnicity were seen at study baseline, changing immigration policy and negative rhetoric did not appear to impact use of pediatric subspecialty care.

**Supplementary File**. The variable "Distance from the clinic" has been rounded to the nearest 10 km.

**Funding:** The authors received no specific funding for this work.

**Competing interests:** The authors have declared that no competing interests exist.

## Introduction

Children of Hispanic/Latino/Latino ethnicity (subsequently referred to as Latino) account for at least 25% of all children in the United States (US). There are an estimated 4.5 million children in mixed status families, in which at least one family member is a US citizen and at least one family member does not have legal immigration status. It is common, for example, for children to be US-born citizens and parents and/or other household members to be a combination of legal residents, undocumented, or naturalized citizens [1, 2]. The Latino population is also one of the largest and fastest growing populations among the major US ethnic groups. In recent years, and particularly since the 2016 Presidential election, anti-immigrant sentiment has become widespread, and immigration policy in the US has become increasingly restrictive. Latino communities have been targeted by immigration enforcement and other law enforcement activities that have been described as biased, punitive, and potentially even illegal [3–5]. While immigration law in the U.S. is primarily enforced by federal agencies, the 1996 addition of section 287(g) to the Immigration and Nationality Act enabled collaboration between federal U.S. Immigration Customs and Enforcement (ICE) and state and local law enforcement agencies, so that local officials could enforce federal immigration laws [6]. Since its inception, the 287(g) program has been repeatedly criticized for racial profiling and civil rights violations due to targeting of immigrants based on appearance, names, or minor violations, with local detainment ultimately leading to deportation for thousands of people. Most recently, in August 2022 the United Nations Committee on the Elimination of Racial Discrimination (CERD) highlighted the harms of the program with regards to harmful, illegal, and ineffective racial profiling, and urged U.S. officials to end the 287(g) program altogether [7]. While only 34 local agencies were participating in the 287(g) program by the end of the Obama administration in 2014, the Trump administration expanded 287g via a Federal Executive Order in early 2017. This action led to a 5-fold increase in program expansion across the U.S., with 140 agencies now participating [8, 9].

Local actions and policies have substantial impact. In North Carolina, a state in the southeastern US with nearly 11 million people, 10.2% of who are Latino [10] there has been considerable controversy over varying participation in the 287(g) program by local law enforcement agencies across the state. In 2018–19 state level policy makers passed a law that would have required all local law enforcement agencies across the state to coordinate with immigration officials and detain individuals in custody for unrelated matters for evaluation of immigration status [11]. While this law was ultimately vetoed by the NC governor, several local agencies within the state chose to enter into new agreements with ICE in order to participate in the 287(g) program. Fear and mistrust due in part to increased immigration enforcement and discrimination may reduce the use of health and social services among Latino families [12–14]. For example, according to a survey conducted in 2019, over 8% of Latino adults in immigrant families have avoided seeking health care because they did not want to be asked about citizenship status [15]. The negative effects of immigration enforcement may also spill over to Latino children who have legal residency or citizenship in the U.S [16]. In a cohort of U.S.-born Latino adolescents, anxiety increased and health status worsened after the 2016 Presidential election, in which anti-immigrant rhetoric played a major part [17]. Previous studies in the US have indicated that states with more restrictive immigration policy had increased prevalence of self-reported poor mental well-being amongst the Latino population when compared to states with less restrictive laws [18]. Additionally, laws enabling local officials to coordinate with ICE previously implemented in other US states were associated with decreased utilization of healthcare resources amongst Latino people not seen in their non-Latino counterparts [19, 20].

For Latino children with chronic health conditions, avoidance of routine health care can result in significant negative health consequences such as disease progression, avoidable use of

acute health care services, and overall increased costs of care [21]. In our academic medical center, serving as the sole source of care for many pediatric subspecialties in a 29-county region in the southeastern U.S., pediatric providers anecdotally noticed decreasing attendance at clinic visits among Latino children receiving specialty outpatient care after the 2016 election. We suspected this decline was related to highly publicized increased local immigration enforcement in early 2017 and worsening anti-immigrant rhetoric and discrimination in areas within and surrounding our service area [22–24]. Therefore, we aimed to describe how clinic attendance of Latino children with chronic health conditions who had previously established longitudinal subspecialty care changed after the 2016 election. We hypothesized that visit cancellations and no-shows would be more common for Latino as compared to non-Latino children followed in our pediatric specialty care clinics.

## Methods

### Participants and data collection

A university-based health system in eastern NC includes academic medical center-based clinics that offer comprehensive multi-disciplinary consultations and treatment for children and adolescents and serves patients referred from more than 145 primary care practices across eastern North Carolina. A single cadre of medical interpreters provides Spanish language interpreter services at all six of the clinics in this study. The business organization that administers the school of medicine clinics at the university has a financial assistance policy for guarantors with annual household income less than 300% of the federal poverty level (FPL) that provides adjustment off charges in the amount of 50%, 70% and 90% for guarantors with household incomes of 201% to 300% FPL, 101% to 200% of FPL and < 100% of FPL, respectively. These discounts apply to Evaluation and Management Services, Diagnostic Procedures and Treatment services that are deemed non-elective and medically necessary. The total FPL discount must not exceed 5% of a clinical department's total charges. Uninsured patients that do not qualify for the FPL discount above are eligible for 30% adjustment off charge amount for Evaluation and Management Services, Diagnostic Procedures and Treatment services that are deemed non-elective and medically necessary outpatient care. There are no prohibitions against providing care for undocumented migrants and/or non-insured patients. We retrospectively queried the electronic medical record (EMR) for children 1–17 years of age who were seen at least three times in 2015 in one of the following ECU clinics: the Healthy Weight (HW) clinic, an interdisciplinary clinic treating children with obesity and associated co-morbidities, the pediatric Cardiology clinic at the East Carolina Heart Institute (ECHI), or pediatric subspecialty clinics operating out of the Pediatric Specialty Clinic (PSC; i.e., Pulmonology, Endocrinology, Nephrology, and Gastroenterology). Together these clinics offer treatment for many chronic diseases, both common and rare, affecting children including obesity, hypertension, kidney disease, congenital and acquired heart diseases, asthma and other pulmonary diseases, apnea and other sleep disturbances, type 1 and type 2 diabetes mellitus and other endocrine problems, chronic constipation, celiac and inflammatory bowel diseases, gastro esophageal reflux, and liver and gall bladder diseases.

At least three visits to the same clinic during 2015 were required, in order to limit the study population to patients with chronic conditions who had successfully established longitudinal care within a subspecialty clinic. Data prior to 2015 were not included due to incomplete implementation of the EMR across the various subspecialty clinics during that period. Based on their most recent visit in 2015, children were stratified according to Latino vs. non-Latino ethnicity, as reported in the EMR from the data entered at the time of initial patient registration. Attendance at follow-up visits was ascertained up to December 2019. We excluded

children with unknown race/ethnicity at the time of their 2015 visits, as well as children with no scheduled follow-up visits after 2015. We also excluded cases with unknown distance to the clinic, or distance >200 km, from analyses incorporating this variable. Other study variables did not have missing data among eligible patients.

## Measures

The primary outcome was visit attendance at any of the ECU pediatric specialty care clinics during 2016–2019, defined as attending the visit, cancelling the visit (including rescheduling the visit, if initiated by the patient), or missing the visit (no-show). Each scheduled visit during this period counted as one observation. In the EMR, clinic visits were originally coded as completed, cancelled, or missed. Scheduled visits were excluded from the analysis if they were cancelled by the clinic, e.g., due to a conflict with another appointment, data entry error, change in provider scheduling, adverse weather, or patient hospitalization, illness, or death. Covariates for our analysis were queried during the most recent 2015 pediatric specialty clinic visit, and included age, sex, race, primary payor (classified as Medicaid vs. other), types of specialty clinics visited during 2015, number of specialty clinic visits scheduled during 2015 (as an indicator of the complexity of subspecialty care required), proportion of specialty clinic visits cancelled by the patient or missed during 2015, and distance to the hospital (based on the postal code of residence) [25–27]. Primary diagnoses were not captured separately in our analysis, due to collinearity with the type of clinic(s) visited during 2015. Similarly, language was not included as a covariate due to collinearity with ethnicity.

## Analysis

Data were summarized using medians with interquartile ranges (IQR) or counts with percentages. For each patient, we calculated the proportion of cancelled visits and no-show visits out of all scheduled visits during the 2016–2019 follow-up period. We compared patient characteristics (at the 2015 baseline) according to whether they cancelled or did not show to any visits in subsequent years using Chi square tests, Fisher's exact tests, or rank-sum tests, as appropriate. For the primary outcome (visit attendance, cancellation, or no-show), we analyzed the visit-level data set using generalized structural equation modeling with a multinomial logit link function. Independent variables in this analysis included the year of each scheduled visit, patient ethnicity, and other covariates described above.

We included an interaction term between year and ethnicity, to determine if the trend in follow-up visit non-attendance differed between Latino and non-Latino patients. This interaction model assumed a gradually growing disparity in visit cancellation or no-show according to patient ethnicity, rather than an immediate, one-time change as might be assumed by a traditional difference-in-difference model. A random intercept at the patient level was included to account for between-patient variability in the likelihood of cancelling or not showing up to scheduled visits, which was not explained by the observed covariates in the model. We used cluster-robust standard errors to account for non-independence of visit outcomes observed for the same patient. Data analysis was performed in Stata/IC 16.1 (College Station, TX: Stata-Corp, LP), and two-tailed P<0.05 was considered statistically significant.

## Ethics

**Ethics approval.** This research study was conducted retrospectively from data obtained for clinical purposes and received ethical approval by the University Medical Center Institutional Review Board (UMCIRB #19–002768).

**Consent to participate.** This study received ethical approval with waiver of individual consent. Consent for publication was not obtained since individual consent was waived.

## Results

We identified 949 patients aged 1–17 years who visited one of the subspecialty clinics at least three times in 2015. We excluded three patients with unknown race/ethnicity, and 94 patients with no further scheduled visits in 2016–2019. The remaining sample of 852 patients (median age: 10 years; IQR: 6, 14; 50% male) included 111 Latino patients, 340 non-Latino White patients, 367 non-Latino Black patients, and 34 patients of another race or ethnicity. During 2015, 67% were covered with Medicaid insurance (the joint federal and state health insurance program in the U.S. for some people of limited income or resources), 29% of patients had been seen in the Endocrinology clinic; 22% had been seen in the Cardiology clinic; 22% had been seen in the Nephrology clinic; 20% had been seen in the Gastroenterology clinic; 20% had been seen in the Pulmonary Medicine clinic; and 13% had been seen in the Healthy Weight clinic (with some patients followed by multiple subspecialists, these percentages add up to more than 100%) Table 1.

**Table 1. Patient characteristics in 2015 by pediatric specialty visit attendance during 2016–2019 (N = 852 patients).**

| Variable | Total (N = 852) | Patients who attended all 2016–2019 visits (N = 110) | Patients who cancelled or no-showed at least one visit in 2016–2019 (N = 742) | P |
|---|---|---|---|---|
| | N (%) or median (IQR) | N (%) or median (IQR) | N (%) or median (IQR) | |
| Age (yr) | 10 (6, 14) | 9 (4, 15) | 11 (6, 14) | 0.312 |
| Sex | | | | 0.664 |
| Female | 427 (50%) | 53 (48%) | 374 (50%) | |
| Male | 425 (50%) | 57 (52%) | 368 (50%) | |
| Race/ethnicity | | | | 0.006 |
| Latino[a] | 111 (13%) | 15 (14%) | 96 (13%) | |
| Non-Latino White | 340 (40%) | 59 (54%) | 281 (34%) | |
| Non-Latino Black | 367 (43%) | 31 (28%) | 336 (45%) | |
| Other | 34 (4%) | 5 (5%) | 29 (4%) | |
| Health insurance | | | | 0.193 |
| Medicaid | 573 (67%) | 68 (62%) | 505 (68%) | |
| Other | 279 (33%) | 42 (38%) | 237 (32%) | |
| Specialty clinics attended[b] | | | | |
| Cardiology | 190 (22%) | 30 (27%) | 160 (22%) | 0.179 |
| Healthy Weight | 113 (13%) | 10 (9%) | 103 (14%) | 0.167 |
| Endocrinology | 249 (29%) | 24 (22%) | 225 (30%) | 0.067 |
| Gastroenterology | 170 (20%) | 23 (21%) | 147 (20%) | 0.788 |
| Nephrology | 185 (22%) | 14 (13%) | 171 (23%) | 0.014 |
| Pulmonary Medicine | 171 (20%) | 22 (20%) | 149 (20%) | 0.984 |
| Percent cancelled or no-show visits in 2015 | 22% (0, 33%) | 15% (0, 25%) | 25% (0, 35%) | <0.001 |
| Total completed visits in 2015 | 4 (3, 5) | 3 (3, 4) | 4 (3, 5) | 0.003 |
| Distance from clinic (km)[c] | 51 (30, 69) | 52 (22, 69) | 51 (30, 69) | 0.840 |

[a] Terminology in the EMR is "Hispanic/Latino"

[b] Clinics attended by the patient in 2015. Categories are not mutually exclusive.

[c] Based on patient postal code of residence. Eighteen patients with unknown distance or distance >200 km excluded.

EMR, electronic medical record; IQR, interquartile range

In 2016–2019, patients in this sample had a total of 9,468 visits, of which 3,104 (33%) were either cancelled (n = 1,945; 20.5%) or for which the patient did not show for the appointment (n = 1,159; 12.2%). Of the 852 patients, 742 (87%) had at least one cancelled or no-show visit during the 2016–2019 follow-up period. Patient characteristics as of 2015 are compared according to this summary measure of 2016–2019 visit attendance in Table 1. In this unadjusted analysis, Non-Latino Black patients and patients seen in the Nephrology clinic were more likely to cancel or no show for at least one visit during 2016–2019. Furthermore, a higher proportion of visits missed during 2015 and a lower number of completed visits in 2015 were both associated with greater likelihood of cancelling or not showing to at least one visit in 2016–2019. However, we saw no difference in the proportion of Latino patients among patients who completed all visits (15/110 patients, 14%), vs. among patients who missed or cancelled at least one visit during 2016–2019 (96/742 patients, 13%).

In the multivariable analysis, we used data from 9,346 visits among 834 patients, excluding 18 patients with missing or implausible (>200 km) distance from the clinic. As shown in Table 2, Latino patients were less likely than non-Latino White or Black patients to cancel a given visit in 2016–2019 rather than complete a visit, (relative risk ratio [RRR] comparing visit cancellation among White patients to Latino patients: 1.41; 95% confidence interval [CI]: 1.08, 1.83; p = 0.012; RRR comparing Black to Latino: 1.56; 95% CI: 1.22, 2.01; p<0.001). However, Latino patients were more likely than non-Latino White patients to not show to a visit rather than complete a visit, (RRR comparing visit no-show among White patients to Latino patients: 0.58; 95% CI: 0.40, 0.85; p = 0.005); there was no difference comparing Black to Latino patients (RRR = 1.09 (95% CI: 0.79, 1.51), p = 0.583).

Other factors associated with increased relative risk of visit cancellation compared to visit completion included having a prior completed appointment in the Healthy Weight, Gastroenterology, or Nephrology clinics during 2015; having missed a larger proportion of subspecialty clinic visits in 2015; and having fewer total completed subspecialty clinic visits in 2015. Similarly, patients seen in the Gastroenterology clinic during 2015, patients who missed a higher proportion of subspecialty visits in 2015, and patients who had fewer completed subspecialty clinic visits in 2015 were more likely not to show for any given subspecialty clinic appointment in 2016–2019. Patients with Medicaid insurance also had a higher relative risk of no-show for visits during the follow-up period (RRR: 1.69; 95% CI: 1.27, 2.26; p<0.001).

Considering trends in follow-up visit attendance, Table 2 shows that in the overall sample, the relative risk of cancelling rather than completing a particular follow-up visit in 2016–2019 was not associated with visit year (RRR: 0.96; 95% CI: 0.91, 1.01; p = 0.125). Likewise, the relative risk of not showing to a visit, rather than completing the visit, did not increase in more recent years (RRR: 0.98; 95% CI: 0.91, 1.05; p = 0.578). We sought to determine if trends in visit attendance were different between Latino patients and patients of other racial/ethnic groups. As shown in Table 3, interacting visit year with race/ethnicity revealed no differences in the trend among these groups. Specifically, among Latino patients, visit year was not associated with either cancelling a visit (RRR: 1.02; 95% CI: 0.86, 1.17; p = 0.972) or not showing to a scheduled visit (RRR: 0.91; 95% CI: 0.75, 1.09; p = 0.307). Predicted probabilities of visit cancellation and visit no-show by year and race/ethnicity are shown in Fig 1, holding all covariates constant at their means.

## Discussion

Our study sought to determine if restrictive immigration policies after the 2016 Presidential election were associated with decreased pediatric subspecialty clinic attendance among patients of Latino ethnicity. Previous studies suggest that restrictive immigration policies,

**Table 2. Multivariable multinomial logistic regression of pediatric specialty clinic visit outcome during 2016–2019 (N = 9,346 visits).**

| Variable | Risk of visit cancellation (vs. completing visit) | | Risk of visit no-show (vs. completing visit) | |
|---|---|---|---|---|
| | RRR (95% CI) | P | RRR (95% CI) | P |
| Race/ethnicity | | | | |
| Latino | Ref. | | Ref. | |
| Non-Latino White | 1.41 (1.08, 1.83) | 0.012 | 0.58 (0.40, 0.85) | 0.005 |
| Non-Latino Black | 1.56 (1.22, 2.01) | <0.001 | 1.09 (0.79, 1.51) | 0.583 |
| Other | 1.35 (0.95, 1.93) | 0.097 | 0.79 (0.38, 1.62) | 0.515 |
| Year of scheduled visit | 0.96 (0.91, 1.01) | 0.125 | 0.98 (0.91, 1.05) | 0.578 |
| Age (yr) | 1.01 (0.99, 1.02) | 0.355 | 1.01 (0.98, 1.03) | 0.456 |
| Sex | | | | |
| Female | Ref. | | Ref. | |
| Male | 0.93 (0.81, 1.06) | 0.241 | 1.01 (0.81, 1.27) | 0.910 |
| Health insurance | | | | |
| Medicaid | 1.01 (0.87, 1.18) | 0.862 | 1.69 (1.27, 2.26) | <0.001 |
| Other | Ref. | | Ref. | |
| Specialty clinics attended[a] | | | | |
| Cardiology | | | | |
| Did not attend visits | Ref. | | Ref. | |
| Attended 1+ visits | 0.98 (0.82, 1.16) | 0.793 | 0.89 (0.64, 1.23) | 0.476 |
| Healthy Weight | | | | |
| Did not attend visits | Ref. | | Ref. | |
| Attended 1+ visits | 1.57 (1.24, 1.99) | <0.001 | 1.32 (0.90 1.93) | 0.151 |
| Endocrinology | | | | |
| Did not attend visits | Ref. | | Ref. | |
| Attended 1+ visits | 1.07 (0.91, 1.26) | 0.408 | 0.86 (0.65, 1.15) | 0.321 |
| Gastroenterology | | | | |
| Did not attend visits | Ref. | | Ref. | |
| Attended 1+ visits | 1.27 (1.04, 1.55) | 0.017 | 1.74 (1.27, 2.40) | 0.001 |
| Nephrology | | | | |
| Did not attend visits | Ref. | | Ref. | |
| Attended 1+ visits | 1.31 (1.08, 1.59) | 0.007 | 1.16 (0.83, 1.62) | 0.398 |
| Pulmonary Medicine | | | | |
| Did not attend visits | Ref. | | Ref. | |
| Attended 1+ visits | 1.06 (0.86, 1.29) | 0.599 | 1.15 (0.82, 1.62) | 0.406 |
| Percent cancelled or no-show visits in 2015 | 1.01 (1.01, 1.01) | <0.001 | 1.01 (1.01, 1.02) | <0.001 |
| Total completed visits in 2015 | 0.95 (0.91, 0.99) | 0.012 | 0.94 (0.89, 0.99) | 0.032 |
| Distance from clinic (10 km)[b] | 1.02 (1.00, 1.04) | 0.055 | 0.98 (0.94, 1.01) | 0.206 |

[a] Clinics attended by the patient in 2015. Categories are not mutually exclusive.

[b] Based on patient postal code of residence. RRR indicates a change in the relative risk for a 10km increase in distance.

CI, confidence interval; RRR, relative risk ratio

increased discrimination, and anti-immigrant sentiment led to decreased health care use among Latino patients and increased self-reported psychological stress in the Latino community [12–20]. Among the pediatric subspecialty clinics associated with our academic health center, we found no change in clinic attendance trends among Latino children in the four years after the election. Although we did not identify a decline in subspecialty clinic use as hypothesized, our study revealed interesting distinctions between missing and cancelling

**Table 3. Multivariable multinomial logistic regression of pediatric specialty clinic visit outcome during 2016–2019, interacting visit year and race/ethnicity (N = 9,346 visits).**

| Variable | Risk of visit cancellation (vs. completing visit) | | Risk of visit no-show (vs. completing visit) | |
|---|---|---|---|---|
| | RRR (95% CI) | P | RRR (95% CI) | P |
| Race/ethnicity | | | | |
| Latino | Ref. | | Ref. | |
| Non-Latino White | 1.71 (1.12, 2.60) | 0.013 | 0.54 (0.31, 0.93) | 0.025 |
| Non-Latino Black | 1.62 (1.08, 2.41) | 0.019 | 0.87 (0.53, 1.43) | 0.584 |
| Other | 1.52 (0.71, 3.22) | 0.278 | 0.62 (0.27, 1.47) | 0.279 |
| Year of scheduled visit[a] | 1.00 (0.86, 1.17) | 0.972 | 0.91 (0.75, 1.09) | 0.307 |
| Year x Race/ethnicity | | | | |
| Year x Latino | Ref. | | Ref. | |
| Year x Non-Latino White | 0.91 (0.77, 1.09) | 0.305 | 1.04 (0.83, 1.32) | 0.721 |
| Year x Non-Latino Black | 0.99 (0.84, 1.16) | 0.858 | 1.12 (0.91, 1.39) | 0.294 |
| Year x Other | 0.95 (0.69, 1.30) | 0.738 | 1.12 (0.81, 1.55) | 0.482 |
| Age (yr) | 1.01 (0.99, 1.02) | 0.349 | 1.01 (0.98, 1.03) | 0.466 |
| Sex | | | | |
| Female | Ref. | | Ref. | |
| Male | 0.92 (0.81, 1.06) | 0.242 | 1.01 (0.80, 1.27) | 0.920 |
| Health insurance | | | | |
| Medicaid | 1.01 (0.87, 1.18) | 0.881 | 1.69 (1.26, 2.26) | <0.001 |
| Other | Ref. | | Ref. | |
| Specialty clinics attended[b] | | | | |
| Cardiology | | | | |
| Did not attend visits | Ref. | | Ref. | |
| Attended 1+ visits | 0.97 (0.82, 1.16) | 0.771 | 0.88 (0.64, 1.22) | 0.460 |
| Healthy Weight | | | | |
| Did not attend visits | Ref. | | Ref. | |
| Attended 1+ visits | 1.57 (1.24, 2.00) | <0.001 | 1.31 (0.90, 1.91) | 0.161 |
| Endocrinology | | | | |
| Did not attend visits | Ref. | | Ref. | |
| Attended 1+ visits | 1.07 (0.91, 1.26) | 0.387 | 0.86 (0.65, 1.15) | 0.319 |
| Gastroenterology | | | | |
| Did not attend visits | Ref. | | Ref. | |
| Attended 1+ visits | 1.27 (1.04, 1.54) | 0.019 | 1.74 (1.26, 2.39) | 0.001 |
| Nephrology | | | | |
| Did not attend visits | Ref. | | Ref. | |
| Attended 1+ visits | 1.31 (1.08, 1.58) | 0.007 | 1.15 (0.82, 1.61) | 0.411 |
| Pulmonary Medicine | | | | |
| Did not attend visits | Ref | | Ref. | |
| Attended 1+ visits | 1.05 (0.86, 1.29) | 0.608 | 1.15 (0.82, 1.61) | 0.416 |
| Percent cancelled or no-show visits in 2015 | 1.01 (1.01, 1.01) | <0.001 | 1.01 (1.01, 1.02) | <0.001 |
| Total completed visits in 2015 | 0.95 (0.92, 0.99) | 0.012 | 0.94 (0.89, 0.99) | 0.032 |
| Distance from clinic (10 km)[c] | 1.02 (1.00, 1.04) | 0.053 | 0.98 (0.94, 1.01) | 0.202 |

[a] The year RRR represents the association between year and relative risk of missing a given visit for Latino patients. This association is modified for other racial/ethnic groups through multiplying the RRR by the corresponding interaction term.

[b] Clinics attended by the patient in 2015. Categories are not mutually exclusive.

[c] Based on patient postal code of residence. RRR indicates a change in the relative risk for a 10km increase in distance.

CI, confidence interval; RRR, relative risk ratio

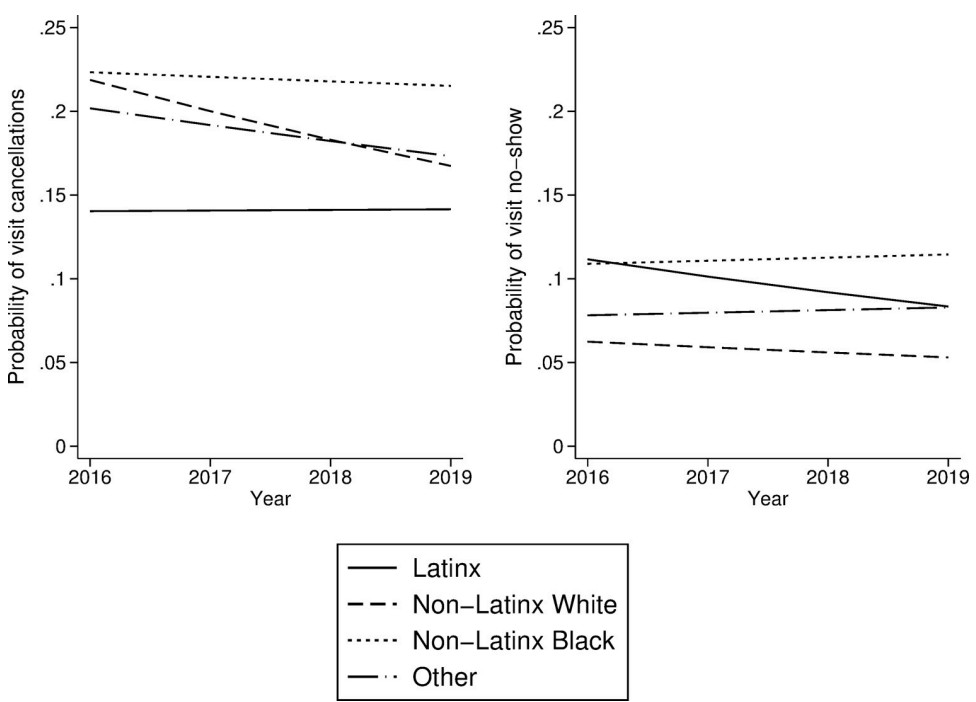

**Fig 1. Predicted probabilities of visit cancellation and visit no-show at pediatric specialty clinics, by year and race/ethnicity, based on multivariable multinomial regression with all covariates held constant at their means.**

appointments as related to patient ethnicity. Latino patients were less likely to cancel visits (contact the clinic ahead of time to cancel a visit) when compared to all non-Latino patients; but were more likely than non-Latino White patients to miss appointments (not show to attend a scheduled visit, without notice). Further work, especially utilizing qualitative methods, is needed to understand distinct factors that increase the risk of missing versus cancelling scheduled subspecialty clinic visits for Latino children.

There are several reasons visit completion may not have changed among Latino patients cared for in pediatric subspecialty clinics over the study period. It is possible that Latino families see physicians and healthcare workers as trusted figures and that people who would be otherwise fearful of authority may still trust physicians; or that healthcare settings could be perceived as "safe" spaces where they would not have to worry about immigration enforcement, although the local clinics did not provide explicit messaging about this. It is also possible that parents feel that the necessity of obtaining healthcare for their children outweighs fears of immigration enforcement, as has been self-reported by parents previously [28]. Additionally, families who fear attending clinic based on immigration status may have had these same fears prior to 2016. This may be more likely to be the case in our study, as North Carolina had the highest number of 287g agreements of any state leading up to 2016; and in subsequent years, several local agencies pulled out of 287g agreements [29]. Additionally, we focused on a patient population seeking subspecialty care. Because the health consequences of missing scheduled subspecialty care may be substantially greater than those associated with missing routine preventative clinical care it is possible that the study population would be less likely to miss scheduled subspecialty appointments compared to routine primary care appointments [26, 28]. Hence our results may not be generalizable to overall health care utilization by Latino children in the US, and this study cannot rule out a negative impact of restrictive immigration policies or negative rhetoric on Latino families' overall utilization of healthcare.

Consistent with our results, previous studies of pediatric outpatient clinic visit attendance found that missed appointments were more common among patients with public insurance [16, 30–39], longer time between referral and appointment scheduling [31–34, 40], and lower socioeconomic status (e.g., measured by lower median income in the postal code of residence) [31, 32, 36–39]. Studies have described mixed results around the associations of patient age, gender, distance from the clinic, and clinic type on visit attendance [30, 31, 33–36, 38–40]. Here we show that distance from the clinic had no effect on clinic attendance, but that the specialty care type of clinic did. We found that a prior completed appointment in the Healthy Weight, Gastroenterology or Nephrology clinics was associated with an increased risk of visit cancellation. Because obesity is a common underlying risk factor for hypertension, non-alcoholic fatty liver disease, dyslipidemia and apnea, conditions that often result in referrals to Nephrology, Gastroenterology, Cardiology and Pulmonology, respectively, patients followed by two or more clinics may feel they are receiving adequate care for multiple chronic conditions at a single subspeciality and hence be less likely to complete all appointments. In a previous study of children with complex chronic conditions treated at our hospital and scheduled for follow-up in affiliated outpatient clinics, we likewise found no difference in clinic visit attendance according to distance from the main campus of the health system [41].

Importantly, prior studies seldom distinguished between visit non-attendance due to cancellation, as compared to non-attendance due to no-shows. In our study, several factors were associated with missed appointments due to both cancellation and no-show, including having missed visits in previous years, having attended fewer clinic visits in previous years, and having attended visits in the Gastroenterology clinic. Attendance in the Healthy Weight and Nephrology clinics was associated with higher likelihood of missing appointments due to cancellation, while Latino ethnicity or Medicaid insurance was associated with a higher likelihood of missing appointments due to no-show. Non-attendance due to cancellation–as opposed to no-show—may offer clinics the opportunity to schedule other patients into the cancelled slots thus maximizing service provided and clinical efficiency. Hence it is particularly important for future studies to elucidate predictors of non-attendance due to no-show in order to mitigate the causes with the aims of maximizing clinical productivity and provision of services.

The conclusions of our study are limited by several aspects of the study design. As a retrospective study utilizing EMR data, our analysis did not have detailed information on families' reasons for missing or cancelling their appointments and whether immigration status or race influenced attendance. We also did not identify immigration status of patients or which patients lived in households of mixed immigration status. While our study involved multiple subspecialities, it was limited in geographic area and may not be representative of clinic attendance in urban centers or areas considered "established gateways" for Latino migration [42]. Among our patient population, we also chose to focus on children who were most likely to require ongoing subspecialty follow-up (based on the criterion of three encounters in the baseline year of 2015), because we anticipated the adverse consequences of visit non-attendance would be highest in this group. However, these inclusion criteria also selected for a population very likely to attend visits despite any fears of immigration concerns, as this group was able to overcome barriers prior to 2016 when local immigration enforcement was substantial. Therefore, families with the most significant barriers around immigration enforcement may not have met criteria for inclusion in the study. Additionally, for children who neither need nor are under ongoing subspecialty follow-up, risk of missing or cancelling a scheduled visit may appear much lower to families so concerns about immigration law enforcement may be weighed more heavily in attendance decisions. Hence our results may not be generalizable to general patient populations.

## Conclusions

We sought to examine whether increased immigration enforcement and rising anti-immigrant sentiment in the US after the 2016 presidential election was associated with subspecialty clinic attendance in Latino vs Non-Latino pediatric patients in a state in the southeastern US. Overall, we did not find a drop off in visit attendance during the four-year period that followed the 2016 election, which may suggest prioritization of pediatric subspecialty care despite immigration fears. However, we did find that when comparing missed appointments due to cancellation vs no-show, Latino patients were more likely to no-show and less likely to cancel ahead of time. The reasons for this are not known but may have to do with phone support with interpreter services or increased social needs such as lack of transportation. Additional research is needed to understand how to reduce the rate of nonattendance, particularly due to no-shows, with the aim of maximizing clinical efficiency and providing scarce subspecialty services.

## Supporting information

**S1 Data. Deidentified final dataset.**
(CSV)

**S1 Code. Code for replicating analyses based on deidentified final dataset.**
(TXT)

## Author Contributions

**Conceptualization:** Dmitry Tumin, Rushina Cholera, Suzanne Lazorick.

**Data curation:** Philip Maness, Suzanne Lazorick.

**Formal analysis:** Dmitry Tumin.

**Methodology:** Dmitry Tumin, Rushina Cholera, Suzanne Lazorick.

**Project administration:** Suzanne Lazorick.

**Supervision:** Dmitry Tumin, Suzanne Lazorick.

**Writing – original draft:** Philip Maness.

**Writing – review & editing:** Dmitry Tumin, Rushina Cholera, David N. Collier, Luisa Bonilla-Hernandez, Suzanne Lazorick.

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
