## [Decision Letter · Decision Letter 0]

10 Nov 2022

PGPH-D-22-01347

Ethnicity and trends in pediatric specialty care clinic attendance at an academic medical center in the rural southeastern US

Dear Dr. Lazorick,

Thank you for submitting your manuscript to PLOS Global Public Health. After careful consideration, we feel that it has merit but does not fully meet PLOS Global Public Health’s publication criteria as it currently stands. Therefore, we invite you to submit a revised version of the manuscript that addresses the points raised during the review process.

You will find Reviewer comments below. Please pay particular attention to Reviewer 2's suggested changes. In addition I have a few requests:

Please add additional context about your clinic and its care and cost policies/practices for undocumented migrants / non-insured.Please revise for a global public health audience, paying particular attention to US-centric and medical language throughout.Please ensure all tables and figures have sufficient information to be understood at a glance.Please provide a visual representation of attended/non-attended visits by year and ethnicity as per Reviewer 2 suggestion.

We look forward to receiving your revised manuscript.

Kind regards,

Hamish R Graham

Academic Editor

Journal Requirements:

1. In the online submission form, you indicated that "Identified study data cannot be released to protect patient confidentiality. Deidentified data are available from the authors upon request.". All PLOS journals now require all data underlying the findings described in their manuscript to be freely available to other researchers, either 1. In a public repository, 2. Within the manuscript itself, or 3. Uploaded as supplementary information.

Additional Editor Comments (if provided):

Reviewers' comments:

Reviewer's Responses to Questions

**Comments to the Author**

1. Does this manuscript meet PLOS Global Public Health’s publication criteria? Is the manuscript technically sound, and do the data support the conclusions? The manuscript must describe methodologically and ethically rigorous research with conclusions that are appropriately drawn based on the data presented.

Reviewer #1: Yes

Reviewer #2: Partly

2. Has the statistical analysis been performed appropriately and rigorously?

Reviewer #1: Yes

Reviewer #2: I don't know

3. Have the authors made all data underlying the findings in their manuscript fully available (please refer to the Data Availability Statement at the start of the manuscript PDF file)?

Reviewer #1: Yes

Reviewer #2: No

4. Is the manuscript presented in an intelligible fashion and written in standard English?

Reviewer #1: Yes

Reviewer #2: Yes

5. Review Comments to the Author

Reviewer #1: The authors attempt to test the hypothesis that increasingly restrictive US immigration policies (and their enforcement) may have impacted access of Latinx pediatric subspecialty clinic patients at their centre. While they identified no significant decrease in clinic visits, there are interesting findings here.

Also, the study team identifies that there are notable design challenges in this study, as their study focuses on a specific clinical centre, on children with chronic conditions (and established therapeutic relationships with their providers), relating to subspecialty pediatric care. Despite these limitations, the study does elucideate novel information about this specific patient population and their healthcare resource utilization, along with the comparisions drawn between Latinx and non-Latinx White patients' clinic interactions via no-shows and cancelled appointments.

Reviewer #2: This is a thoughtful analysis asking a question of interest to child health professionals, policy makers, and human rights advocates in the United States. While I am not an expert on multinomial regression, the authors carefully describe their model, e.g., use of a random intercept at the patient level given that their visit-level outcomes are not independent of one another. I also appreciate the differentiation between canceled visits and "no-shows" (is there a better term, e.g., failure to arrive?).

There are a few areas for revision.

First, the overall narrative is US-oriented. Brief explanations of terminology like "Latinx" or noting that North Carolina is a US state would ensure the writing is easier to follow for a global audience. As a minor related point, the authors describe the Latinx population as the fastest growing ethnic group in the US, but I believe this is actually people of Asian origin.

Similarly, I wonder whether a public health audience will understand all the nuances pertaining to pediatric specialty care. Broadly, the article seems to be written with clinician researchers in mind, rather than a public health audience. What is "Healthy Weight" for example? More to the point, does it matter when considering the main hypothesis of the study? Perhaps some of those details either need to have more context or to move to a supplement for clinicians?

Third, the authors seem to assume that all Latino/Latinx families in their sample are of immigrant or mixed status. (The latter is jargon, by the way, and merits explanation in the text.) It is likely they have firsthand experience in the community or with local demographic data, and so for the authors this assumption may be reasonable. As a reader, however, it remains a major methodological question: Perhaps the results are null because the analysis focuses on a largely non-immigrant, non "mixed status" community who have no immigration-related fears.

Fourth, some of the tables may be missing helpful information, e.g., the referent group for some variables?

Finally, it would be nice to see a visual representation of no-shows by race/ethnicity by year. Interaction terms are notoriously difficult to interpret.

6. PLOS authors have the option to publish the peer review history of their article (what does this mean?). If published, this will include your full peer review and any attached files.

**Do you want your identity to be public for this peer review?** For information about this choice, including consent withdrawal, please see our Privacy Policy.

Reviewer #1: No

Reviewer #2: No

---

## [Decision Letter · Decision Letter 1]

21 Mar 2023

Ethnicity and trends in pediatric specialty care clinic attendance at an academic medical center in the rural southeastern US

PGPH-D-22-01347R1

Dear Dr Lazorick,

We are pleased to inform you that your manuscript 'Ethnicity and trends in pediatric specialty care clinic attendance at an academic medical center in the rural southeastern US' has been provisionally accepted for publication in PLOS Global Public Health.

Best regards,

Julia Robinson

Executive Editor

Reviewer Comments (if any, and for reference):

Reviewer's Responses to Questions

**Comments to the Author**

1. If the authors have adequately addressed your comments raised in a previous round of review and you feel that this manuscript is now acceptable for publication, you may indicate that here to bypass the “Comments to the Author” section, enter your conflict of interest statement in the “Confidential to Editor” section, and submit your "Accept" recommendation.

Reviewer #1: All comments have been addressed

2. Does this manuscript meet PLOS Global Public Health’s publication criteria? Is the manuscript technically sound, and do the data support the conclusions? The manuscript must describe methodologically and ethically rigorous research with conclusions that are appropriately drawn based on the data presented.

Reviewer #1: Yes

3. Has the statistical analysis been performed appropriately and rigorously?

Reviewer #1: Yes

4. Have the authors made all data underlying the findings in their manuscript fully available (please refer to the Data Availability Statement at the start of the manuscript PDF file)?

Reviewer #1: Yes

5. Is the manuscript presented in an intelligible fashion and written in standard English?

Reviewer #1: Yes

6. Review Comments to the Author

Reviewer #1: none

7. PLOS authors have the option to publish the peer review history of their article (what does this mean?). If published, this will include your full peer review and any attached files.

**Do you want your identity to be public for this peer review?** For information about this choice, including consent withdrawal, please see our Privacy Policy.

Reviewer #1: No
